# Practice of Simulation and Life Cycle Assessment in Tribology—A Review

**DOI:** 10.3390/ma13163489

**Published:** 2020-08-07

**Authors:** Abdulaziz Kurdi, Nahla Alhazmi, Hatem Alhazmi, Thamer Tabbakh

**Affiliations:** 1National Center for Building and Construction Technology, King Abdulaziz City for Science and Technology, P.O. Box 6086, Riyadh 11442, Saudi Arabia; akurdi@kacst.edu.sa; 2Material Science Institute, King Abdulaziz City for Science and Technology, P.O. Box 6086, Riyadh 11442, Saudi Arabia; nalhazmi@kacst.edu.sa; 3National Center for Composite and High-Performance Materials, Center of Excellence for Aeronautics, King Abdulaziz City for Science and Technology, P.O. Box 6086, Riyadh 11442, Saudi Arabia; 4National Center for Environmental Technology, King Abdulaziz City for Science and Technology, P.O. Box 6086, Riyadh 11442, Saudi Arabia; halhazmi@kacst.edu.sa

**Keywords:** tribology, simulation, life cycle assessment, friction and wear

## Abstract

To simulate today’s complex tribo-contact scenarios, a methodological breakdown of a complex design problem into simpler sub-problems is essential to achieve acceptable simulation outcomes. This also helps to manage iterative, hierarchical systems within given computational power. In this paper, the authors reviewed recent trends of simulation practices in tribology to model tribo-contact scenario and life cycle assessment (LCA) with the help of simulation. With the advancement of modern computers and computing power, increasing effort has been given towards simulation, which not only saves time and resources but also provides meaningful results. Having said that, like every other technique, simulation has some inherent limitations which need to be considered during practice. Keeping this in mind, the pros and cons of both physical experiments and simulation approaches are reviewed together with their interdependency and how one approach can benefit the other. Various simulation techniques are outlined with a focus on machine learning which will dominate simulation approaches in the future. In addition, simulation of tribo-contacts across different length scales and lubrication conditions is discussed in detail. An extension of the simulation approach, together with experimental data, can lead towards LCA of components which will provide us with a better understanding of the efficient usage of limited resources and conservation of both energy and resources.

## 1. Introduction

Tribology is the division of science that deals with rubbing surfaces that are in relative motion. As a result of relative motion between involved surfaces, frictional force arises and subsequently “wear” takes place. These friction and wear cause excessive energy consumption as well as loss of materials in the form of wear debris [1]. The word “tribology” (Greek term “τρίβο”, meaning “to rub”) was first coined by physicist David Tabor in 1964 [2] and globally introduced and defined by H.P. Jost in 1966 [3]. Abrasion, shearing, and adhesion are three different phenomena that govern friction between rubbing surfaces [4] and result in material removal from one or both mating surfaces. Unlike friction, wear is a system response, as it is influenced by changing input parameters as well as motion conditions, environmental and material parameters of a tribo-system [4]. Typically, major issues in tribology deals with the nature of the surface (i.e., roughness) and contacts, frictions and energy loss, wear and material loss and lubrication conditions. In plain language, tribology investigates friction and wear behavior of materials under various loading and motion conditions. The findings from these investigations help to optimize both operation parameters and design of material system for energy efficient operation of the system. Such an efficient system not only helps to conserve energy but also help to reduce global warming. The dominant cause of global warming are the emissions from the use of non-renewable fossil fuels which supply approximately 80% of the world’s energy in terms of electricity, heat, transportation. and manufacturing from plastics to heavy metals [5]. The interrelated challenges of excessive consumption of fossil fuels, limited supplies of non-renewable energy sources, greenhouse gas emission, energy shortages, and global warming clearly suggest that focus should be given especially to these two issues:Increasing the overall energy efficiency; andReducing the carbon footprint by all possible means from any active engineering systems.

These two issues are particularly applicable to the transportation and manufacturing industries compared to all other sectors [6]. For example, case studies regarding worldwide use of passenger cars and paper production machines indicate that, passenger cars consume 33% of their total fuel energy per year in order to overcome friction, whereas paper production machines use 32% of the electrical energy to overcome friction [6]. Globally, 100 million TJ is spent per year for this frictional loss, which contributes towards harmful greenhouse emission with 7000 million tons of CO_2_ [6,7]. Thus, investigation of tribological aspects to minimize friction and wear is a never-ending quest. As new alloys/materials are being used in different components, detailed know-how on their tribological properties are not unquestionable. The tribological optimization of any system can be achieved in several ways such as:Use of lubricants (both inorganic and organic) in order to prevent direct surface contact between mating components [8];Application of suitable tribological coatings (including engineered soft coatings and solid lubricants) on the mating components [9]; andCombined application of tribological coatings on mating components with presence of lubricants within the same tribo-system [8].

These pathways undoubtedly lead to significant heat reduction generated due to the friction; therefore, savings of a significant amount of energy and materials are foreseen. From these aspects, tribology is capable enough to meet the demands of a sustainable society.

Traditionally, experimental approaches are utilized to investigate tribological behavior of materials in question. Based on experimental data, the wear rate of materials is presented, together with their frictional behavior [10]. In addition to that, prevailing wear mechanisms can also be unrevealed by post investigation of wear tracks together with the evolution of coefficient of friction over time. The most common apparatus towards that is pin-on-disk instrument [11]. In addition, different forms of fretting instruments can also be used [12]. It is also a common practice to attach climate chambers with such instruments to investigate the effect of test environments (i.e., humidity, inert gas, vacuum condition) on friction and wear [13]. In addition to experimental approaches, simulation-based techniques are also used to investigated friction and wear [14]. Usually, simulation technique requires considerable time to compute a given problem, depending on complexity. As better computing power is available, it became efficient to run a simulation to investigate friction and wear behavior of a system to obtain a quick output. This is one of the inherent reasons for the recent surge in the use of simulation techniques [15]. There are quite a number of simulation techniques available to investigate the tribological aspects of tribo-system [16]. Though most of the techniques are capable of adequately handling the basic question of friction and wear, a handful of them can address complex scenarios. Selection of any particular technique depends on a number of factors:

Capability of the technique to handle the problem in question;Availability of required resources (computing power) and expertise; andVerification of the simulation results.

Different researchers use different techniques based on the abovementioned criteria and came out with different outcomes, which is sometimes conflicting. Moreover, inappropriate selection of simulation techniques and parameters (i.e., nodes, sub-cluster, elements) not only provides wrong results but also put the validity of the technique in question. In addition to the above mentioned physical and simulation experiments, life cycle assessment (LCA) of the tribological system has received lots of attention recently from the scientific community [17,18]. The drive behind that is to maximize the use of a given tribological system from an environmental point of view by analyzing the use of each component, including lubrication from “cradle to grave”.

In view of that, the objective of the present paper is to make a comprehensive review of different simulation techniques used in tribology and how it can be extended towards the life cycle assessment of components. This review will help the researcher to have a quick overlook towards the selection of the most appropriate techniques to answer their research query and help the engineers to achieve the best practice in real life components. To accomplish the objective, a systematic approach was taken. The manuscript starts with a brief comparison between physical experiment and simulation approach in tribology. After that, different modelling and simulation techniques were introduced with primary focus on machine and deep learning process. Fundamental of any simulation approach is based on solving contact mechanics equations for a given situation. Development of equations representing contact mechanism are well-established in literature dating back to 1805 by Young [19]. Later on, Hertz equations on contact of elastic solids solely dominated the development of tribological-related equations [19]. As these equations were lengthy in nature and easily available in the literature, it was not discussed in this paper. The current work mainly focuses on machine learning and deep learning-based simulation techniques that are less covered in literature [15,20]. Towards that, effect of lubrication on the simulation of tribo-contacts is put in focus, which will help to understand the phenomenon of wear across length scale from the nano- to macro. A snap shot of the limitations of current simulation practice in tribology is then discussed together with an example in industrial applications. Lastly, life cycle assessment (LCA) in tribology is discussed in view of the interrelation among physical experiments and simulations. The manuscript is wrapped up with the conclusion and a number of broad directions for future work that are imperative to enhance our current understanding of friction and wear through simulation practice. Thus, the general aim of the manuscript is to demonstrate the role of simulation practice to address tribological-related application and how it can be extended towards LCA of the tribological system.

## 2. Physical Experiment versus Simulation Approach in Tribology

Usually, before deploying components in service, lab-scale tests are carried out to predict the lifetime of such components in services. Different labs (i.e., research groups) approach this in different way depending on the availability of resources and expertise [21,22]. In most cases, it is based on their respective experiences, best practices, and also on customer’s demands. However, to achieve most realistic outcomes, a field survey is carried out prior to any lab tests to find out the exact operating conditions and environments [23]. Based on field survey data, the most appropriate input parameters are selected to conduct lab-tests. Based on the results of lab-scale tests, components may be improved further by playing with different materials and coatings in hand [24]. Though lab-scale tests are most widely used and offer reliable results, it is not free from some flip sides [25]. For examples, lab-scale tests are expensive to carry out as it requires both equipment, experimental setup, and manpower. Frequently, lab-scale tests are carried out in accelerated mode and small-scale which overlook both time-effect and size-effect, respectively. To overcome such limitations, computer-based simulations were introduced. Computer-based simulations overcome couple of limitations of lab-scale experiments at the expense of huge computing capacity and requires excellent know-how of the physical process, which is not readily available. Nevertheless, computer-based simulation provides some unique aspects that are not easily attainable by experimental techniques. For example, it provides realistic estimation on energy consumption based on lubrication, wear, and friction concepts on contact surfaces like bearings and pistons [5]. It plays an important role in projects to carry out the work in a given time frame and budget limit.

### 2.1. Benefits of Physical Experiment

A physical experiment refers to the approach of deducing hypotheses tested in a given survey and to analyze the data to draw an inevitable conclusion of a particular problem. Tribological testing is designed and carried out to meet a defined need, and it is important as it reveals vital information regarding any failure mechanisms of mechanical components. Some of these experiments include the effects of surface roughness which helps to understand surface-related technology [26]. Therefore, it becomes easier to predict and determine the kind of rough surface required for a given contact surfaces. Additionally, physical experiments help to identify a material/material system that are appropriate to be used in a given system with increased efficiency. Physical experiments enable us to measure friction and wear between the surfaces to be able to predict the requirements for the system to function accordingly [27]. In this case, it becomes easier to understand the exerting high pressure at contact surfaces. The knowledge of high pressure is essential in the manufacturing process of lubricants and hydraulic fluids for enhanced performance in a machine. Last but not the least, physical experiments are beneficial in tribology for helping to identify activities at the molecular level. As a result, it becomes possible to determine the exact thickness of lubrication films that needs to be in place to reduce energy conservation. Therefore, the experimental concept in tribology is inevitable to optimize the parameters necessary for increased industry performance [27]. These parameters include the material system and kinds of lubricants to be used in different contact surfaces.

### 2.2. Benefits of Simulation Approach

Simulation, in recent years, plays a vital role to predict the behavior of components in service as well as the life cycle assessment (LCA) of the components. The applications of simulation to improve productivity can lead to significant financial savings. Additionally, the results will be more authoritative and cover all possible combinations of parameters, which is not practically attainable in an experimental approach. For instance, simulation aids in manufacturing lubricants used in systems such as bearings, shafts, and gearboxes by applying molecular simulation concepts by taking consideration of the behaviors of molecules under high pressure [28]. Simulation also enhances hydrodynamic performance in tribo-contact, and different simulation methods, like multi-phase oil-gas simulation, are useful to reduce mist formation at contact surfaces such as crankshafts. Such mist formation impacts on the operations of the system negatively by reducing the availability of lubricant in confined areas [29]. Precisely, simulation is beneficial to mimic tribological contacts as it is capable of positively impacting and applying solutions in industrial quires. Furthermore, the concept applies to areas like nano-mechanics and microelectronics which involve understanding the molecular flow of lubricants. For instance, through simulation, it is easier to apply the concepts of elastic, kinetic, and thermal interaction in industrial manufacturing process [28]. Therefore, simulation benefits industries by ensuring that there is low energy consumption caused by friction and as a result, the system can perform effectively. Examples towards that include conversion of energy in hydraulic motor to minimize friction and power required in the operations, improving productivity, efficiency, and quality of products. It can be gained by finding ways to deal with various industrial issues which have affected the productivity of manufacturing systems.

### 2.3. Interdependency between Simulation and Physical Experiments

The concept of simulation in tribology is intertwine with actual phenomenon taking place in tribological systems. Thus, each individual approach benefits from the other. Simulation helps to identify the areas that require physical experimentation to increase system efficiency. On the other hand, physical experiments provide details such as oil measurements to be used in simulation process of manufacturing lubricating fluids. As a result of experimentation in tribology, the concept of simulation is applicable effectively to enhance the science of friction and lubrication in machines to increase energy conservation effects. There are a lot of similarities between simulation and physical experiments. However, each of them provides extra information than other. Physical experimentations can provide empirical data to test, but simulation cannot produce such data. Additionally, realistic experiments can offer operations directly in targeted system as an extra ability. Simulation, on the other hand, provide interactive ways to correlate different parameters. It also offers to test unrealistic scenarios of involved parameters that are dangerous to engage without imposing any risk to the operator or instrument. In this case, it becomes easier to understand system functionality thought the real system is unrealistic to build. Such extra information is not attainable from physical experimental approach, as there is a need for a linear system to conduct an experiment. For example, it is impossible to measure turbulence quantity at the nano-meter scale, which is attainable by using a computerized simulation approach. Last but not the least, a simulation approach minimizes errors that happen most in physical experimentation because of operator error or instrument compliance or both [30].

## 3. Different Modelling and Simulation Techniques

The term “modelling” has several definitions depending on classification and the discipline in which this term is used. For example, in computer science the definition of the model, according to Marvin Minsky [31] is as follows: to an observer B, an object A* is a model of an object A to the extent that B can use A* to answer questions that interest him about A. On the other hand, according to Jeff Rothenberg [32], in artificial intelligence, the modelling is defined as: “Modelling in its broadest sense is the cost-effective use of something in place of something else for some purpose”.

A model is a simplification of reality which could be physical or mathematical in nature. A mathematical model represents the system using mathematical equations, while a physical model is simply the reality of splitting the system into elements that relate to each other by a physical relationship. The two main elements in modelling are: (i) variables—the model’s elements in the system; and (ii) parameters—the factors that succeed the variables into equations. There are two types of variables, namely, endogenous and exogenous. The variables that are used in the calculations of the system within the model are called endogenous, while the variables that are calculated outside the system that has been modelled are called exogenous [33]. The appropriate solution, when dealing with modelling whether it is in tribology or any other system, the most important aspect is to understand the limitations and to take all the conservations and limitations into consideration during analyzing the results obtained from modelling. To model a complex and multi-system, such as tribology, researchers need to decompose the system into sub-systems that are easy to handle in simulation package [34]. On the other hand, effect of lubrication, either external or self-lubrication, plays an important role in tribo-contacts that ultimately dictate overall friction and wear of the system. To understand the effect of lubrication in tribo-contacts, computer-based simulations are not only cost effective but also helps to visualize the process at tribo-contacts, which is otherwise not achievable. Such simulations explore the influence of sliding conditions, different geometrical parameters and environmental settings on lubrication, wear rate and coefficient of friction. The main three factors that play significant role in tribological simulations are mechanical properties of the system, relative motion of sliding surfaces, and the way in which resulting heat dissipate from sliding surface [35]. The most commonly-used simulation techniques to model and predict the lubrication effects are machine learning method, molecular dynamic simulation, finite elements and boundary conditions method, crystal plasticity and discrete dislocation dynamics, solid/fluid interactions, and finite volume method [15] as shown in Figure 1 and described in subsequent sections.

### 3.1. Molecular Dynamic Simulation

Molecular dynamic (MD) simulation was initially used in tribology to mimic the scenarios of the interaction of hard spheres [36]. Then, it was extended to investigate other mechanical and physical phenomena [37]. Molecular dynamic is based on calculating the kinematics between particles in molecular level by using Newtonian or Langevin laws. The key factor in MD is the force field or interaction potential which could explain dynamic atomistic method but not the interaction between them. Recently, research in atomic scale friction by using atomic force microscope (AFM)/lateral force microscope (LFM) [38] and surface force technique in nano-scale or atomic-scale or by computational simulations of MD has increased significantly [39]. Although MD simulations are limited to a system size of hundreds of atoms, they provide a fundamental understanding and method to aid missing experimental setup at atomic scale. For example, Harrison et al. [40,41] shows the benefit of using MD simulation on the behavior of solid lubricants for friction of diamond composite film, diamond-like carbon, amorphous carbon, and self-assembled monolayer systems [42] which are used in many applications, such as in space technologies, micro-electromechanical system (MEMS), and hard disk storage [43,44].

Typically, in MD simulations, atoms are presented as discrete particles where their paths are calculated by integrating the equations of motion to obtain velocity, stress and strain, force and other external parameters. After considering their properties, geometrical parameters, and boundary conditions, a set of equations are developed that best represent the scenarios. In addition, temperature differences, heat transfer, and dissipation through the system can also be taken into account due to the presence of external sliding force. Therefore, there are different methods that can be used in simulations to maintain a constant temperature (Langevin dynamics) [42,45]. Langevin dynamics approach was used to describe Brownain motion by introducing an additional term that represents friction in classical dynamic equation [46,47]. To simulate tribo-contact by MD simulation, displacement between counter face and surface are set be a constant value to allow the contacts between two surfaces by applying a normal force or a pressure, as represented schematically in Figure 2. Details of the governing equations of MD simulation is available in literature [48,49,50,51] and thus were not included here.

Typically, surface features can be varied depending on surface roughness and materials’ composition and properties, which need to be analyzed. Usually, the counter face is made of a known material that has a well-defined microstructure and allows a comparison of the results of different simulations. An alternative method is to use same materials for both surfaces and thermostats method could be used [35]. However, if this method is being used, both the position of thermostat and choice of thermostat need to be considered. Otherwise, either a choosing wrong position or parameters can cause a large fluctuated force and results in not enough control over temperature.

In MD simulations, all the information for each atom is stored for every step, such as velocity, force components, acceleration, and positions. In order to interpret the data and transform them into useful information, several key parameters should be monitored such as friction force and normal force between surfaces, displacements between surfaces, structural changes, and dislocations and overall chemical bond breakage/formation between atoms.

### 3.2. Finite Elements and Boundary Elements Method

There are number of numerical methods that have been developed to simulate physical phenomena in mechanics. The main techniques that have been used ultimately are finite element method (FEM) and the boundary element method (BEM). Typically, finite element method sets an explicit relationship between stress and strain with a finite strain formulation [52]. While BEM considers the relationship between force and pressure with displacement in two orthogonal directions [53].

Typically, frictional and adhesive problems are considered to be nonlinear for two contact surfaces due to the unknown contact area [15]. To solve that, one of the approaches is to couple FEM with BEM to achieve a comprehensive solution or to use discrete element method. To apply FEM and BEM on tribological problem, geometry of contacted bodies needs to be discretized into small elements followed by imposing boundary, initial conditions, and materials’ properties according to problem specification. In fretting fatigue and wear analysis, mesh needs to be fine enough so that it can capture the stress field near the edge. This methodology is illustrated in Figure 3.

To describe the contact between two bodies theoretically, imagine that there are two bodies, namely, body A and body B body, with a small gap between them as shown in Figure 4. There are two methods that can explain the contact between these two bodies, which are the Lagrange multiplier method [54] and penalty method [55]. The governing equations of FEM simulation were developed based on the Lagrange multiplier method [56] and are available in literature. To simulate wear by using FEM, energy approach is used, which assumes that wear can be achieved by using local energy used in the wear process, as described by Fouvry et al. [57].

Application of FEM/DEM to answer the questions related to tribology is widely available in literature. For example, Ducobu et al. [58] developed a FEM to investigate the effect of geometry on tool wear that was made with Ti6Al4V [58]. Perazzo et al. [59] used BEM to predict abrasive wear on steel against various types of copper ore [59]. Rojas et al. [60] applied BEM to investigate the effect of friction on relative wear on mining hopper and their results showed that the model was sensitive to grain shape and better verification with experiments was achieved when clusters were used instead of spheres [60]. Zhang et al. [61] developed FEM to predict fretting fatigue and wear and analyzed the effect of stress distribution. The results showed that, fretting crack at the edge was repressed by fretting wear; while fretting crack at inner surface was endorsed by stress concentration that occurs near the edge of fretted scare [8]. Krop et al. [62] developed a numerical simulation to investigate scratching on polymer composite which was polycarbonates [62], while Gao et al. [40] used the same technique on talc-filled polypropylene polymer. Lian et al. [63] used FEM to model fatigue life and wear on railways by applying equivalent pressure and heat source on rail surfaces. The fatigue life was found to be decreased with slip ratio. Din et al. [64] predicted the damage on fiber-reinforced polymers caused by adhesive wear by using FEM in various in-plane directions of sliding relative to fibers. Lone et al. [20] presented a state-of-the-art review on the applications of the extended finite element method (XFEM) which is useful for studying many engineering discontinuities phenomena such as frictional contact. The main functions that covered XFEM are Heaviside jump functions, Lagrangian multiplier, and penalty approaches [20].

### 3.3. Discrete Dislocation Dynamics (DDD)

Discrete dislocation dynamics (DDD) method is used to model the plasticity phenomenon at micro-scale level by modelling solid material as a linear elastic continuum where the dislocations are the means of linear elastic fields [65,66,67,68]. The boundary value is solved as in terms of superposition’s principles by using the fact that both dislocations and sloid material are defined as a linear elastic continuum. The boundary value solution is obtained at each time interval and at each material point as a whole of dislocation fields and their image fields [15]. The image fields can then be calculated by using finite elements or by using Green’s function molecular dynamics (GFMD) [69].

### 3.4. Crystal Plasticity

Crystal plasticity is used to model elasto-plastic deformation of metal, in which it is assumed that plastic deformation is a result of plastic slip-on metal crystals [70,71]. This is due to the presence of shear stress on crystallographic slip system which beats the corresponding critically resolved shear stress and theoretically could be expressed as a slip rate. Crystal plasticity has been proven to be an efficient method for modelling elastoplastic phenomenon of polycrystalline aggregates [72,73]. This method has been extended to include deformation twinning and martensitic transformation deformations [74,75,76]. In addition, visco-plastic self-consistent (VPSC) model is used to model the hardening and plastic forming process [77].

### 3.5. Finite Volume Method (FVM) and Solid/Fluid Interactions

All the modelling techniques mentioned previously (molecular dynamic simulation, finite elements method, boundary conditions method, crystal plasticity and discrete dislocation dynamics method) were focused on modelling sliding behavior between two dry surfaces [15]. However, in some cases, solid/fluid interactions should be taken into account when modelling lubrication effect in the presence of a fluid film between two contact surfaces. This technique involves the simulation of complex interactions among particles. Unfortunately, the majority of modelling researchers have ignored solid/fluid interactions for simplicity sake, and in order to avoid expensive computational power [15]. However, finite volume method (FVM) has been developed to deal with this complex domain using mass conservation method [78,79,80]. The development of a solver that deals with solid/fluid interactions by taking into account multiphase phenomena and complex geometry is still in its development stage. This method involves using fluid flow conservation equation through discretized geometry by the solution of Reynolds equation with a mass cavitation which could be applied in lubrication process [81]. Finite volume method is also applied with regard to lubricated contacts where the film thickness was discontinuous and concentrated inertia effect in the Bernoulli equation has been taken into account [82].

### 3.6. Machine Learning

Machine learning (ML) is a relatively new approach that can perform complex pattern recognition and regression analysis in a valid way [83]. This can be done without constructing and solving physical models. Among various ML algorithms, artificial neural networks (ANNs) are widely used due to the availability of large data sets together with sophisticated algorithm architecture. Though ANNs are considered as a phenomenological method, a well-trained ANN can include mechanistic understanding of the considered problem at hand. Machine learning (ML) is the procedure of developing a model that finds the patterns in a set of data by training the algorithm by given data set without human involvement. After that, the model can be used to predict new data as schematically shown in Figure 5.

There are several machine learning techniques that could be used, depending on the type of dataset that has been collected. If the dataset does not have any output data, the technique is called unsupervised machine learning, and the only way that this data can be dealt with is by clustering, where the data is grouped by finding the hidden pattern in the data [84]. On the other hand, if the input and responding output data are available, machine learning is referred to as supervised machine learning and can be used to train the algorithms to predict the response to any new unseen data. Supervised machine learning can take the form of classification models that classify and separate input data into classes, categories or groups. The second type of supervised machine learning is that of regression [84]. This is used when the responding output is a real number, such as when predicting temperature or the lifetime of equipment or, as in this case, for predicting the friction coefficient or wear rate in tribology. The regression process is used as a Gaussian process, linear or support vector regression or regression trees [85,86]. It is worth noting that ANNs can be used for both regression and classification problems. This lifecycle is illustrated in Figure 6.

To have a successful machine learning model, the process must pass through a lifecycle step. Whether it is a regression problem or classification problem, the main steps start with data acquisition and end with the evaluation and verification of the model. These steps are summarized in Figure 7. After collecting the data, specific features must be chosen by selecting the most important and vital features that influence the results and their weights. Then, the developer has to select a model to train these data sets and learn from given data. Finally, the last step involves estimating statistical errors by comparing predicted results against real results. Some important features of ML steps are discussed in subsequent sections.

There are a variety of machine learning algorithms that are applied to resolve tribological-related problems. However, every method has its own application area which is difficult to be generalized for all other applications. Data quality, amount of data, operating and system conditions affect significantly the performance of the model and relative error in predicted data. There are several ANN methods reported literature such as radial basis function (RBF) neural network [87], probability neural network (PNN) [88], Bayesian network (BN) [61], and support vector machine (SVM) [89] to diagnose wear scenarios in engines. In addition, back propagation (BP) neural network has been used to detect wear-related faults in piston rings [90]. Anand et al. [16] developed a model to establish the input/output relationship of friction welding using ANN. Then, this model was used for optimization of friction parameters by using a force ANN technique [91]. Zakaulla et al. [92] predicted coefficient of friction and wear rate of polycarbonate-based composite by using ANN and the main feature parameters were testing conditions and composition of the materials [92]. Borjali et al. [93] employed different machine learning algorithms, such as gradient boosting, M5m CART, and linear regression, to predict the wear rate of polyethylene quantitively from pin-on-disc experiments [93]. Xu et al. [94] developed three data-driven models, which are ANN, belief rule base (BRB) and evidential reasoning (ER) from the dataset that was generated from a wear-related problems in a diesel engine. The results showed that the system was enhanced by improving fault tolerant ability [94]. Wang et al. [95] improved the funnel sliding mode control for servo devices after replacing the scaling factor that results from tracking errors estimated from NNs [96]. Gouarir et al. [97] used convolutional neural network (CNN) to predict tool wear by using a training dataset from dry machining with a non-coated ball on a stainless steel surface. Results were about 90% accurate compared with experimental measurements. Vaira and Padmanaban [98] developed a neural network (NN) that can predict the tensile strength of aluminum alloy by knowing rotation speed, shoulder, and pin diameters. Their model predicted the tensile strength that was close to experimental data. Tran et al. [85] predicted wear by using Gaussian process regression in a very short time and a small dataset (i.e., 144 training points and 20 testing points). The data that were used in regression was generated from a steady-state computational fluid dynamic (CFD) simulation. Slavkovic et al. [99] used the data from wear rate of iron casting to train an SVM and improved support vector machine (ISVM) techniques. The errors of predicted results were 5.85% and 6.67%, respectively [100]. Danaher et al. [101] predicted wear of Ni alloys in high temperatures. Suresh et al. [102] adopted a ANN and applied it to model wear on polymers. As evident from above examples, ML is a popular simulation tool to investigate tribo-contact-related problems and address friction and wear.

#### 3.6.1. Machine Learning Methods

As mentioned in previous section, the ML process effectively make use of two main analysis process, namely, linear regression (LR), regression tree (RT), support vector machine (SVM), decision tree (DT), gaussian process regression (GPR), and artificial neural networks (ANNs) which are described hereafter.

##### Linear Regression (LR)

Linear regression uses the linear relationship between input parameters (independent variables) and output parameters (dependent variables) [103]. Previous studies which employed linear regression in modelling and predicting the wear rate or friction coefficient, indicated that the linear model was not able to predict the coefficient of friction or wear accurately in a tribological system, especially in a complex system which involves many operating conditions and different materials parameters. In contrast to that, nonlinear models work better and were able to predict coefficient friction with an error of about 5% [104]. Archad et al. [105] was one of the first researchers to use the linear model to estimate metal wear [106]. Dan Jia et al. [107] used linear regression to model feature parameters such as molecular energy, low orbital energy, dipole moments and fat–water partition coefficient of the lubricating oil and related wear [107].

##### Regression Tree (RT)

The principle of regression tree is similar to that of decision trees. However, the target variable is in the form of real numbers and can consist of continuous values. Regression tree has been used to predict friction coefficient in ceramic materials. Two matrices have been employed to choose the important attributes which are the measured sum of squares and the variable importance in projection. The final regression tree indicates that the density of ceramic material, melting point, and cation radius have a larger influence on coefficient friction relative to other parameters [108].

##### Support Vector Machine (SVM)

A support vector machine (SVM) is used in many classification applications due to the fact of its accuracy [109]. An SVM is a supervised machine learning technique developed based on statistical learning theory [110], that uses less data than ANN [111]. Support vector machine is a widely used classifier non-parametric model that uses a non-linear method that allows a tolerance in loss function when training the model. It does this by ignoring any loss that is less than a small value that has been pre-determined [109]. This technique has been used in tribological applications to estimate defect size from a limited experimental dataset [112]. The classifier SVM obtains more accurate results in predicting bearing fault detection by using time domain and vibration, different loads and speeds, than classifier ANN.

##### Decision Tree (DT)

The decision tree (DT) is a supervised machine learning technique that uses a hierarchical approach to represent the classification of observation objects by testing variables’ attributes. The main principle of DT is to select the integrity of variables which their attribute represents the most gain value. This attribute will be selected as a root for the decision tree, and every set should contain data of a similar type [113,114]. The procedure will be carried out until the final nodes in a hierarchical representation are obtained. The DT has been used successfully to detect bearing damage and incipient defects at early stage [115].

##### Gaussian Process Regression

Gaussian process regression (GPR) is based on assumption that there is a directional likelihood between input and output for any model of the process. When the input is (*x*) and output is (*y*), the conditional distribution of the probabilities become *p (y/x)* [116]. The regression process by using Gaussian distribution is built on a collection of random values and that every point provided in the model delivers information of the adjacent value afterwards [86,117]. The graphical illustration of GPR is shown in Figure 8.

The box represents observed variables which are uncertainly independent of the other corresponding value of *f*, while the circles show unknown variables [86].

##### Artificial Neural Networks

Artificial neural networks (ANNs) consist of simple artificial neurons that connect with each other by passing information through links between them [118]. Any ANN model could consist of one linear layer or a complex architecture of input layers (deep hidden layers) and finally output layer, as can be seen in Figure 9.

An ANN is a relatively fast simulation process, compared to the MD simulation technique and widely used in regression and classification type problems. Use of deep learning method together with ANN requires a high-quality dataset. The performance of the simulation process becomes more accurate with both the quality and volume of data set compared to classical ML algorithms. A summary of the simulation methods with main features are listed in Table 1.

### 3.7. Data Acquisition and Cleaning Data Set

To prepare the data for machine learning procedure, the dataset should be split into a training set (which should be the maximum percentage of the data), the validation set (which is used to examine whether or not the model can be generalized for any new data), and the test set (which is used to measure the errors in the model and to compare the percentages of the errors with the training set). Attention should also be given to data acquisition process and subsequent cleaning of the acquired data set. It is not advisable to collect all the data (both necessary and unnecessary) what is available from the system as an output [102,121]. Such an approach will not only increase the size of the data set (which causes data storage issues) but will also waste computing time. Thus, before the simulation, it should be critically analyzed which data set should be collected against known experimental conditions. After collecting, the dataset should be validated by identifying linear correlation coefficient among input and output data [37,93]. In addition, data should be analyzed by determining maximum and minimum data value, standard deviation, and stability. This step is crucial to determine the weight and contribution of every data point in output results to avoid underestimation or overestimation of the effect on target data. Sometimes it requires to clean the data set to filter out noise from the system and great care on that should be practice avoiding any over or under cleaning of the data set [102,122]. Whether the data have been collected through physical experiments or by modelling or analytical technique, it should be clean, well-structured, labeled, and augmented in order to be analyzed. This process is known in data science as data preparation. The first step of this process is that of data extraction and profiling. This means extracting raw data from unstructured sources to create a typical balanced and organized profile. The next step is cleaning the data by detecting any missing points. It is also necessary to scan the data to ensure its accuracy and comprehensiveness. Then, if possible, the raw data should be labeled in order to allow the supervised learning to be homogeneous. Finally, the feature engineering step must be taken by choosing the most effective features that influence the data and their weight [123,124].

### 3.8. Input and Output Parameters

Tribology encompasses various subjects such as physics, chemistry, and mathematics. Accordingly, input parameters vary a lot and influence respective output parameters [26,29,30,31]. External loading conditions and geometry, materials’ behavior and system property such as properties of lubrication are some of the input parameters [32]. Output parameters, on the other hand, include coefficient of friction, boundary condition of lubrication and measurement of wear rate [29,30,33,34]. During simulation, a database is used where some or most of the parameters can be retrieved. Such a database is obtained from a literature survey, by computational modelling, or by experimental measurements. The data are related to the material properties of the balls (during ball-on-disk tests) or pins (during pin-on-disk tests) in terms of mechanical hardness, surface roughness, dimensional characteristics, in terms of chemical composition [30], in terms of physical properties (melting point, cation radius [35], viscosity, and density [36]), test conditions, such as different temperature or operating pressures, or different load and speed conditions [37]. Accordingly, the output parameters were related to wear rate or defect size [38]. In addition, in classifier SVM, roughness of the surface was considered as an important output parameter that was used to determine the fault detection of bearings [39]. At the same time, the inner race fault, outer race fault, and conditions parameters can be used to predict fault detection [40]. Other parameters used for classifiers ANN and SVM include time domain, vibration signals, and various load and speed conditions [41].

### 3.9. Model Evaluation

The predicted values from machine learning algorithms are usually validated by calculating statistical errors. These errors are computed by comparing predicted values with respect to real values in order to evaluate the accuracy of proposed models. These are ultimately known as statistical errors in the form of root mean square error (RMSE) (Equation (1)) [125]:(1)$$RMSE=\sqrt{\frac{1}{T}\overset{T}{\underset{t=1}{\sum }}{\left(\hat{y}\left(x_{t}\right)-y\left(x_{t}\right)\right)}^{2}}\;$$
where *t* = 1, 2, 3, …, *T* accounts for the transects, $\hat{y}\left(x_{t}\right)$ is predicted value, and $y\left(x_{t}\right)$ is the actual value. Mean absolute error (MAE) is another form of statistical errors as represented in Equation (2) [126]:(2)$$MAE=\frac{1}{T}\sum_{t=1}^{T}\left|\hat{y}\left(x_{t}\right)-y\left(x_{t}\right)\right|$$

As the model becomes more accurate, predicted values should be close to real values and statistical errors should be close to zero [121].

### 3.10. Simulation across Length—Scale

Pin on disk (POD) and ball on disk (BOD) configurations are the two most widely used tribological test configuration that are reported in the literature and contact size between surfaces are in the range 50 to 800 µm in the macro-scale and between 0.06 and 6 µm in micro-scale [127]. As a rule of thumb, the contact area between tribo-contacts at the micro-scale tribology is about five orders of magnitude lower compared to the macro-scale tribology [127]. This affects the friction forces and surface tension in addition to the influence of contact shape and surface roughness. To overcome the problem of measuring contact area at the atomic level and their unclear boundaries experimentally, researchers have tried to theoretically measure the contact between two bodies at different scales by taking into account the atomic structure and differences between adhesive and non-adhesive contacts [128,129]. It is not possible to verify such contact areas by physical experiments, where the materials are transparent and boundary line is close to non-detectable range [130]. To accurately measure the atomic contact area, it is assumed that the contact area between two bodies is the sum of the areas of their atoms that were in contact [131]. Researchers have used various methods to simulate the materials at atomic scale, such as using the average distance, potential energy, or the distance-based limits for the structure of the materials [132]. However, these methods do not take into consideration some important facts such as the inhomogeneous behavior of multiple elements and alloys or the nanoscale roughness of the materials [15]. Another approach to overcome such complexity is to assume that the discrete surface is a continuous one and to calculate the mean of bicubic splines in Persson’s theory. In addition, in the presence of fluid, it will be difficult to report the viscosity of thin film at nanoscale roughness. Itoh et al. [133] reported the viscosity of perfluoropolyethelene (PFPE) film deposited experimentally on a substrate that was used on hard disks. This viscosity value has since been used to predict the tribology of hard disk interface [133].

Another approach towards that is to carry out physical experiments at different length scales and then co-relate it accordingly to the method proposed by Basak et al. [38]. In their techniques, the authors have used three different tribo-systems dedicated to measuring the co-efficient of friction at the macro-, micro-, and nano-scales, respectively. These emphasize only the difficulties to achieve such outcomes but also the requirements of such state-of-the-art instruments and expertise which are not readily available [134]. In that respect, multi-scale modeling techniques that involve machine learning and deep learning are foreseen to play an important role [94,118,135]. Figure 10 represents the general overview of different length-scale investigation as reported in Reference [127].

## 4. Limitations of Current Simulation Aspects

With the advancement in instrumentation, it is possible to use a surface force technique, such as AFM/LFM, to measure friction and wear at the nano-/atomic scale. However, using a computational technique such as FEM or MD provides more information about initial surface conditions and atomic interactions. However, even with the availability of high computational power, simulations are still challenging and require accurate in put parameters and molecular dimensional sampling. Therefore, coupled modelling and experimental work can provide more accurate results. For example, experimental results from AFM can shows the range of curvature and boundary conditions, followed by verification using FEM or MD simulations. In addition, one of the basic practices that could be used to discrete simulations is to use probability density functions and the likelihood of distribution. This includes any uncertainty that can be accrued in the system, such as uncertainty in input parameters and measurements or structural building of geometry. Therefore, noisy and fluctuated data sometimes needs to be used which might complicate the system [34]. One other approach that could be used to overcome the limitation is to evaluate the results against experimental results when they are available. This technique is ultimately used to evaluate the model and to refine uncertainties in input data or structural uncertainty [33]. There are several limitations which should be considered when modelling tribology aspects whether the modelling is carried out by MD or FEM or ML. These limitations are as follows:

### 4.1. Limitation of the Modelled System

Sometimes, for simplicity or lack of information, the system is divided into subsystems and not modelled as a whole real physical system. This is due to the complexity and composed components in a real world system which is difficult to model for limited computer power or difficulty of capturing the requirements that control the system.

### 4.2. Limitations of Variables and Parameters

Regardless of the application and nature of variables and parameters, all parameters have some level of uncertainties which should be considered when developing an accurate model. Parameters are usually tuned or calibrated when measurements are conducted to ensure fitting of the data and reduce errors when analyzing the results. This type of limitation is sometimes referred as parametric uncertainty. In addition, when using a dataset to train a machine to learn an algorithm, we need a huge dataset to ensure predicted results are accurate, which is expensive in terms of computational power and time or experimental cost; as the data could be collected through some sort of experiment [83]. Currently, to overcome the issues of availability of data, researchers have developed an auto encoder to generate a random dataset that agrees with the pattern of the previous data [136]. One example of a famous auto encoder is the variational auto encoder (VAE), which transforms input data into Gaussian distribution that has a mean, variance, and standard deviation to generate similar new data [137].

### 4.3. Limitation of System Environment

There are some environmental and boundary conditions that are hard to be captured in complete real system. This is not related to computer power resources; however, this is related to the difficulties of capturing full information about the modelled system at every step and validating it against real world conditions which have many assumptions.

### 4.4. Limitations of Data Labelling

Sometimes, because of lack of experience, it is difficult to label data that are used for supervised learning in machine learning. Making the judgments on collected dataset and choosing right label for them need deep knowledge, especially if the machine learning developer is not the same person who collected the data. In addition, choosing effective parameters and balancing them and how much they affect the results is crucial for accurate modelling. These steps in machine learning are called feature engineering and it means the process of selecting most useful and important parameters that influence the predictions.

### 4.5. Overfitting or Underfitting Issues

For a limited dataset, a surplus of input data can cause overfitting of the model which affects the predicted results and increases statistical errors. The overfitting problem means that the model is trying to fit in every data points, even noisy data. On the other hand, underfitting problem means that the model fails to find a pattern in the dataset by ignoring some of the feature data points. There are many techniques that can be used to avoid these vital underfitting and overfitting problems such as regularization and dropout [137,138].

## 5. Industrial Applications

Tribology always has been connected to industrial problems from the beginning [139]. Tribology is an applied science and, as such, many industrial components are designed based on tribological rules which is crucial to modern machinery that involve sliding and rolling surfaces [140]. The interest in advancing the efficiency of interacting surfaces in relative motion is a concern for many companies, particularly those that have contact surfaces, such as bearings, and in the medical practice. Modeling and simulation are more and more upcoming tools in various fields in engineering specific conditions in the field of tribology. However, the benefits of simulation come at the cost of computational power. As shown in Figure 11, computational period increases exponentially with feature size and becomes out of control far before it reaches life-size components.

Simulation is much like running a field test, except that the system of interest is replaced by a physical or computation model [15]. One of the uses of the simulation concept is in the medical practice for hip joint fixing and replacement processes, and a chronological aspect of that is tabulated in Table 2.

These artificial body parts result in friction thus requiring lubrication to prevent wear and increase their time in application. For practitioners to resolve this issue, they utilize simulation concept which enhances the accuracy of the parts to improve their in vivo performance [140]. In addition, simulation application involves approximating and establishing the limit of material used in the manufacturing of prostheses.

Another example of industrial application of simulation is the machines that use crankshaft and piston movement in their production processes including motor vehicles. When the flow of these elements occurs, they initiated the formation of mist, which reduces lubricant availability in the vicinity. Multi-phase simulation of gas–oil is applicable in this respect to resolve flow phenomena [88]. Precisely, in this process, simulation application is to alter the behaviors of lubricant molecules at contact surface under high pressure to influence the tribological processes. In this respect, molecular dynamics simulation helps to clarify the issue for the possibilities of predicting the dynamics of tribological functionality and coefficient of friction [149].

Before industrial revolution, fraction and wear were controlled by applying animal fat or oil. During the industrial revolution, operating conditions at contacting surfaces changed dramatically and [150] simulation played a vital role in the industrial process for designing lubricants to reduce friction and wear, which led to increased efficiency of the system [87,150]. Hydrodynamic lubricants are one method used extensively to support load under reduce fraction [6,7]. The devices are enhanced to act in full film hydrodynamic conditions, which requires a molecular dynamic simulation concept to improve the oxidation stability of the oil [151]. Additionally, finite hydrodynamic bearing issues are some of the problems in the industries resolved by the application of simulation. The solution of the problem is to produce smooth isothermal surfaces by an increase in numerical techniques [139]. It is thus essential in producing thin and thick film lubricants which are useful in increasing energy efficiency in machines as industries can produce lower viscosity lubricants using molecular dynamic simulation. An example of this concept is the toroidal traction drive used in automotive; whose efficiency increased in recent years because of the simulation aspect application to reduce friction in hydraulic systems [152]. The concept has also improved gas turbines and rings of diesel engine piston for automotive to help such systems withstand higher pressure without the effect of oxidation volatilization or degradation through using synthetic lubricants.

## 6. Life Cycle Assessments (LCAs)

For a given tribo-system, there are a number of individual components termed as tribo-elements. Each of the tribo-elements perform their functions in a synchronized way to run the system that involve transfer of energy and material. Transfer of energy and material increases wear and causes poor performance of the system, which can be minimized by applying proper lubrication and/or selection of materials such as coatings. This will, in turn, lead to conservation of material and energy. This integrated approach is known as life cycle assessment (LCA), where the life span of individual tribo-elements is taken into consideration including disposal at the end of service life. Thus, life cycle assessment of any product includes environmental concerns related to extraction of raw materials, manufacturing and usage phases, and, at the end, recycling and disposal of components [134]. A general logical diagram of LCA is shown in Figure 12 that represents the flow of steps considered in LCA.

In LCA of tribology lubrication, energy conservation, environmental conservation, and recycling of tribo-elements form the base of a rectangular pyramid that leads to the conservation of energy at the apex of the pyramid [154,155]. Pyramidical representation of these interrelated factors is also known as life cycle tribology (LCT). Thus, by careful analysis of the lifespan of each tribo-elements, innovative and creative tribo-techniques at designing can be adopted to extend the lifespan of a given tribo-system. The LCA of any tribo-system is directly and indirectly influenced by a number of participating factors such as the wear-resistant behavior of the system, lubricating behavior, extending life span by lubricating and re-furbishing the components [16,91]. These factors are mostly in-built into any tribo-system. The transformation of material and energy consumption during tribological interaction defines the performance and life cycle of that particular system [156]. Several researchers have performed in-depth LCA studies for different tribo-systems, such as:Production of lubricant based on lubricant restoration, where fresh lubricant shows higher impact on LCA compared to restored lubricant [18];Role of tribology in LCA of mechanical systems and development of environment management system (EMS) to support that [157];Role of tribology in LCA of mechanical systems focusing on conservation of energy and material [158];Development and use of environment adapted lubricants (EAL) instead of commercially available inorganic lubricants in order to reduce the impact on LCA [159];LCA of automotive bearing materials and its impact on global warming [160,161];LCA of different natural biodegradable lubricants with reference to commercial lubricants [162];Creating a sustainability pyramid for environmentally friendly hydraulic fluids and comprehensive LCA studies in order to understand the product category within the sustainability pyramid [163]; andTribological durability of a domestic refrigerator, where the refrigerant replacement shows high friction and wear characteristics of tribo-elements leading to higher energy consumption [164].

Beside these, some researchers have also focused on modelling LCA of tribo-elements used in mechanical systems. For example, a tribology-based impact pyramid model demonstrates a useful idea of life-cycle tribology (LCT) [4] or digraph and matrix method model of tribo-elements demonstrate design and developmental needs [154]. There are also tribological research studies, which have concluded that adopting state-of-the-art and creative tribo-techniques during design and functioning stages of a product can significantly improve poor friction and low wear resistance, thereby eventually leading to a reduction of overall energy loss [165,166], for example, use of efficient coating and lubricant in wind turbine components. Among all renewable energy technologies, wind turbines are fast growing. In the past two decades, global wind power capacity has increased from 6100 MW (in 1996) [167] to 591 GW (in 2018) [166]. The operation of wind turbines involves several challenges such as wear of the main shaft, gearbox bearings and gears, electric arcing on generator bearings, erosion of blades via solid particles, presence of cavitation and problem induces by natural resources such as rain and hails [122]. On the other hand, tidal-power turbines, particularly common in Europe and North America [168], face issues such as erosion, lubrication due to the seawater, oils and greases, corrosion, and biofouling [169]. To address the abovementioned issues, nano-structured coating has been developed to increase the life cycle and efficiency of wind turbine components facing friction and wear [170]. As coating was applied on base components; thus, after wearing out the coatings in service, instead of replacing the whole component, it needs recoating of the surfaces [169].

Performance of a tribo-system, in terms of wear and friction, is evidently enhanced by the use of lubrication, which improve overall efficiency of the system. By changing, replenishing, and applying lubrication on a regular basis, maintenance costs could be reduced in noticeable amount. Lifetime lubricants is the best way to reduce their cost. Lifetime lubrication means that that once applied, there is no further need for reapplication of lubricants over the lifetime of the system. Lifetime lubrication can be achieved by improving the levels of additives in the base lubricant stock while still meeting operational requirements [114,166,171]. Aside from improving additive levels, surface coatings, solid lubrication, and self-lubrication methods can be applied to achieve lifetime lubrication [18,155,172]. A number of reports are available in the literature in order to reduce the health risks posed by conventional lubricants by the use of new and innovative methodologies [159]. These studies have shown that, environmental impacts can be reduced by environmental engineers and tribologists through the use of water lubrication, oil-coated water droplets, ceramic/ceramic nitrogen lubrication, surface micro-texturing, and cleaning by di-electrophoresis and electrophoresis. Environmentally acceptable lubricants (EALs) are becoming more accepted as eco-friendly methods of tribo-element lubrication. The aforementioned methods of lubrication not only reduce the health hazards and risks involved with conventional lubrication, but they also have a substantial impact on the LCA of the systems involved [159,160,161,164].

Typically, non-biodegradable lubricants are used in wind turbine gearboxes to prevent the bearings and gears from micro-pitting and erosion. A new ester-free formulated oil-based lubricant on hydro-isomerized API group III oil and a stable shear-resistant poly alkyl-methacrylate (PAMA) thickener has been developed to improve micro-pitting and erosion resistance properties. This newly formulated lubricant meets the requirements for both standard industrial gear oils and component specifications from wind turbine manufacturers [173]. Beside this, investigations have been done in order to minimize wear through application of low friction coatings in the presence of biodegradable lubricants and the tribo-pairs used this system are brass–glycerol–steel. The formation of a low-friction tribo-layer during sliding reduces wear after being transferred to the mating surfaces [174,175]. With the help of simulation of those scenarios, it is possible to estimate the lifetime of such critical components and replace it accordingly before catastrophic failure of the whole structure. With the help of LCA, it will be possible to take it in one step forward on resources and energy conservation.

Minimizing friction and, therefore, wear, has gained a lot of researchers’ attention, as it reduces carbon footprints and greenhouse gas emission with altering normal operational conditions of a running system. This leads to reduced carbon emissions in terms of monetary and ecological factors, which are the key to tribological sustainability. Two decades ago, a study on US energy consumption indicated an approximately of 11% savings due to the developments in lubrication and tribology in four major areas: transportation, power generation, turbo-machinery, and industrial applications [174]. Development of materials with super-low friction, biomimetic materials, biodegradable lubricants, and smart coatings are keys towards sustainability in tribological applications. Sustainable product design models have developed in order to assess environmental, social and economic impact of the product [175]. Further to this, development of environmental conscious materials, such as green waxes, lubricants, and adhesives, has also been reviewed by several researchers from a sustainability point of view [176]. Another aspect is design and development of new materials in order to reduce overall frictional losses due to the heavy-weight machine parts. Several high strength-to-weight ratio materials, such as specific aluminum alloys and aluminum matrix composites, have been developed without compromising tribological aspects particularly for automotive and aerospace applications [177,178]. Thus, the need is always there, where tribologists should work together by combining the resources of experimental, simulation, and LCA towards the growth and implementation of sustainable tribology not only for research purposes, but also from commercial applications point of view.

## 7. Interrelation among Physical Experiments, Simulation, and Life Cycle Assessment

Physical experiment, simulation and life cycle assessment aspect of any tribo-system is interrelated with each other. Thus, the techniques are complementary to each other than competitive. For example, it is difficult to investigate the effect of surface roughness and lubrication on friction and wear at nano-/micro-scales over a wide length [4,38,134,168,179]. In that case, simulation is particularly helpful, as it is easy to vary such input parameters (surface roughness and lubrication) in the system. On contrast, simulation is ineffective at macro-scale investigation of the effect of such parameters on friction and wear, as the required computing power is beyond achievable. Thus, the general approach is to conduct physical experiments according to the Taguchi method [180] and quantify the contribution and interaction among input parameters on output parameters. Based on this, together, with the help of underlying friction and wear mechanisms, a simple equation-based model can be developed. Later on, this initial model can be extended with a comprehensive dataset. If it comes out that the data set is failing at a certain stage of simulation, then additional experiments could be conducted to cover the gap and a new data set can be included in the simulation package. In this feedback and regression way, by combining both physical experiments and simulation approaches, a more accurate simulation process can be developed and integrated into the LCA methodology.

## 8. Conclusions

In this review paper, current simulation approaches to address tribological-related issues was reviewed. As contact mechanism-based approaches are well-established and well-accepted in the literature; this current work mainly focused on machine learning and deep learning-based simulation techniques that have been less covered in the literature. Towards that, the effect of lubrication on the simulation of tribo-contacts were of particular focus which helped to understand the phenomenon of wear across length from the nano- to macro-scale. As evident from above discussion, the simulation approach not only provides certain and accurate specifications of scenarios in hand but also offers a complete picture by combining the interaction of various input parameters that are experimentally unavailable. Together with experimental results, simulation can help for better product design and efficient usages of resources by taking into consideration the life cycle assessment of the components. However, the present simulation approach is limited by the way “uncertainty” is included in the model. Thus, a holistic approach is required among product designers, researchers, and simulation developers to understand the problem in hand, in full scale, and address accordingly. Product developer must consider both the “aleatory” and “epistemic” uncertainty that are involved in the design. It is also required to address the “uncertainty” in the model in such a way that it becomes meaningful beyond model developers. This can offer a dramatic impact on engineering design of the tribo-contact components. Thus, to maintain a clear physical understanding of the outcome of complex models, those processes will preferably be first studied on their own, before introducing related behavior laws in more comprehensive models. In this way, it will be possible to maintain an advanced level of understanding of frictional interfaces and come out with simple but comprehensive models that are capable to optimize and control relevant process parameters. This approach will also ensure that, modeling of such tribo-contact issues is not just kept in academic exercises of computing capabilities, but also comes with a reasonable experimental setup to verify model outcomes. In this context, quantitative comparison with physical experimental results will lead to considering plasticity, long-range adhesion, and large deformations into the model, beyond contact mechanics-based simulation approach.

## 9. Future Work

It is needless to say that the practice of simulation and LCA on tribology has come a long way. Withstanding the capability of the modelling approach to effectively address elastic problems of tribo-contact, a substantial effort is foreseen to take into consideration the effects of plasticity, adhesion, friction, wear, lubrication, and surface chemistry into tribological models. In addition to that, as a number of systems involve two or more of the abovementioned phenomena simultaneously in tribo-contact, multi physics models are challenging to develop. This challenge can only be overcome by a collaborative effort among multidisciplinary expertise.

The next issue that needs to be addressed elaborately is multiscale effect. Due to the deviation from ideal surface (without any roughness) to real surfaces (surface roughness, chemical heterogeneity, deformation of contacts) heterogeneities lead to energy dissipation. This heterogeneity not only imposes a non-linear behavior but also depends on scale size. In addition to that, hierarchical (multi-scale) surface interaction mechanisms may influence friction and wear in a different way depending on the nature of involved surfaces. Thus, the challenge is to cover a whole length scale, from nano- to macro-scale, in a single simulation package. More integrated work is foreseen in that aspect with fast growing computational power and super computers. This does not mean to underestimate the essence of physical experiments, which is always required to validate the simulation outcomes.

## Figures and Tables

**Figure 1 materials-13-03489-f001:**
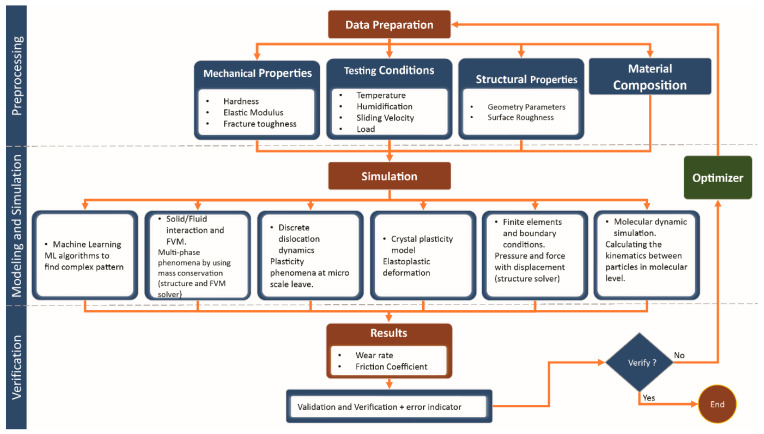
Overview of different simulation techniques commonly used in tribological related applications.

**Figure 2 materials-13-03489-f002:**
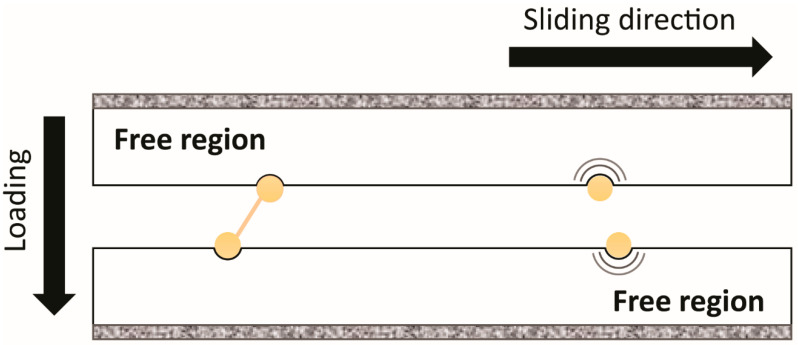
Schematic representation of molecular dynamic simulation of a tribo-contact.

**Figure 3 materials-13-03489-f003:**
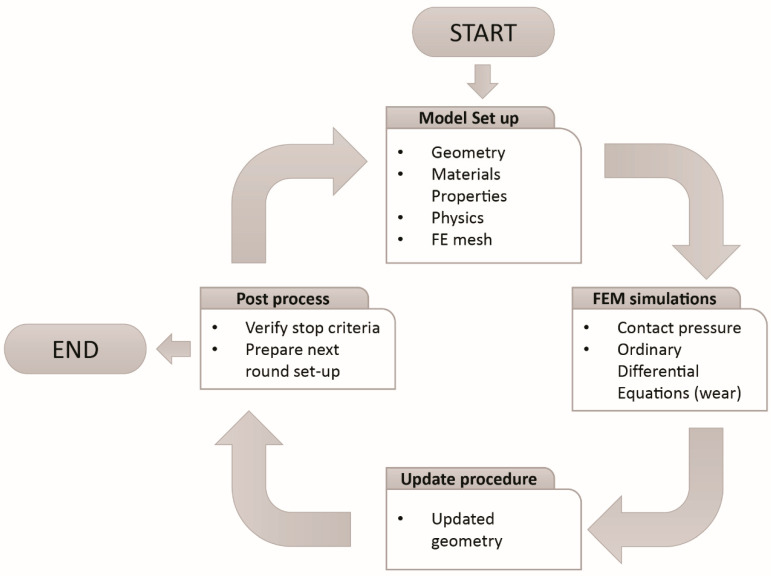
Typical finite element metho and boundary element method simulation steps for tribology.

**Figure 4 materials-13-03489-f004:**
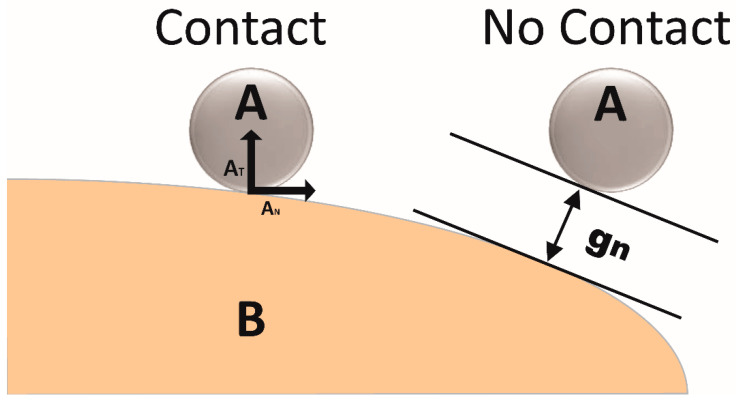
Schematic diagram of sliding contact between two bodies.

**Figure 5 materials-13-03489-f005:**
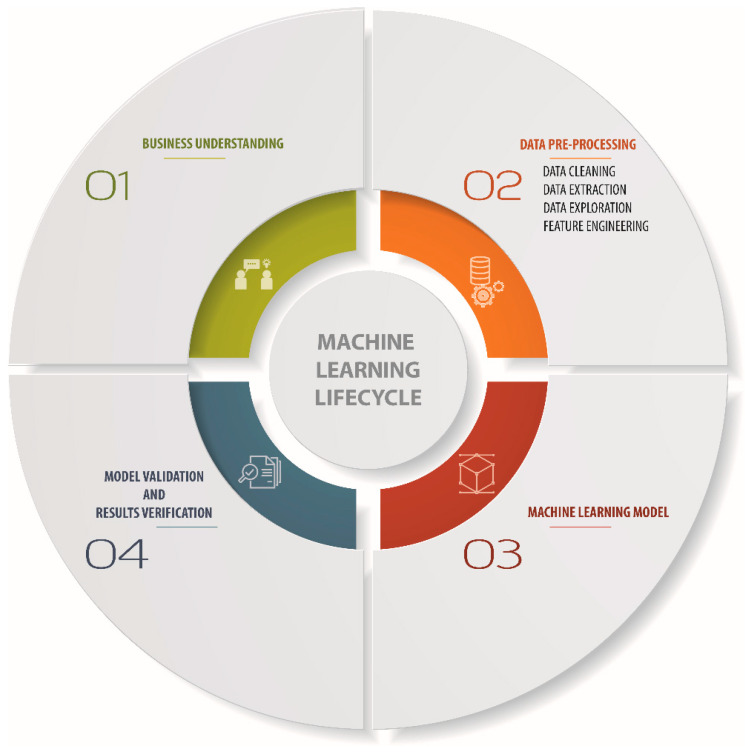
Machine learning lifecycle.

**Figure 6 materials-13-03489-f006:**
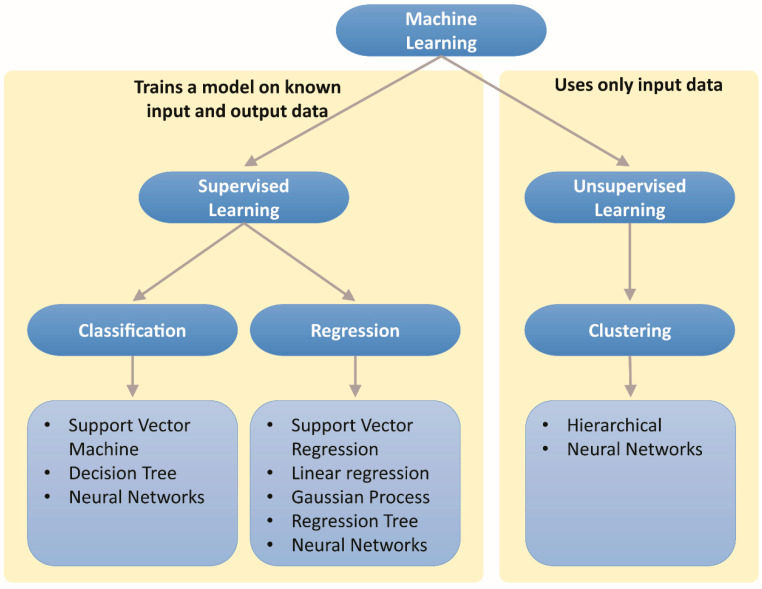
Classification of typical machine learning process.

**Figure 7 materials-13-03489-f007:**
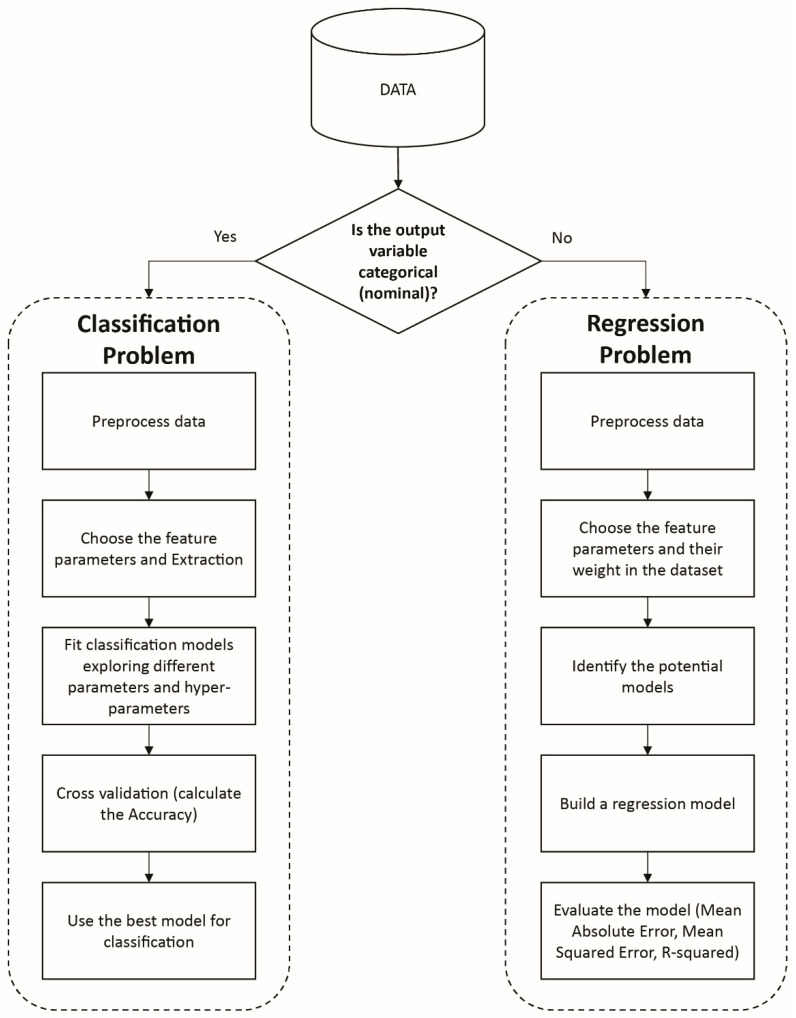
Typical data science main steps in ML.

**Figure 8 materials-13-03489-f008:**
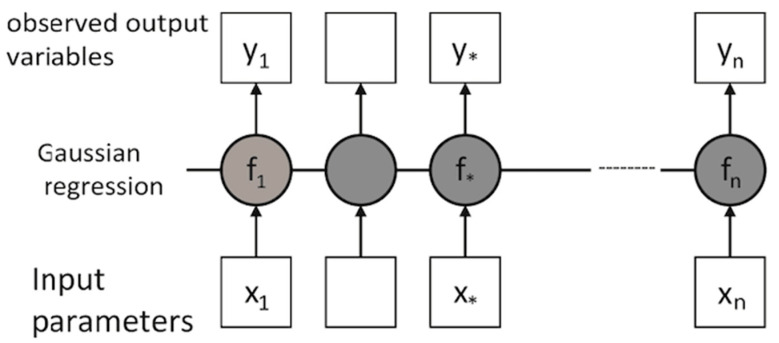
The graphical illustration of GPR (figure redrawn based on discussion in Reference [86]).

**Figure 9 materials-13-03489-f009:**
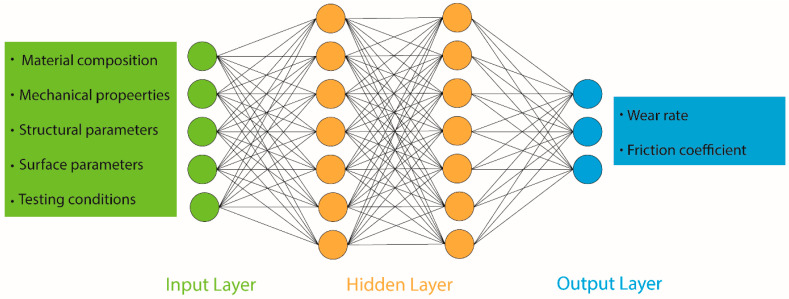
Typical architecture of an artificial neural networks used in tribology.

**Figure 10 materials-13-03489-f010:**
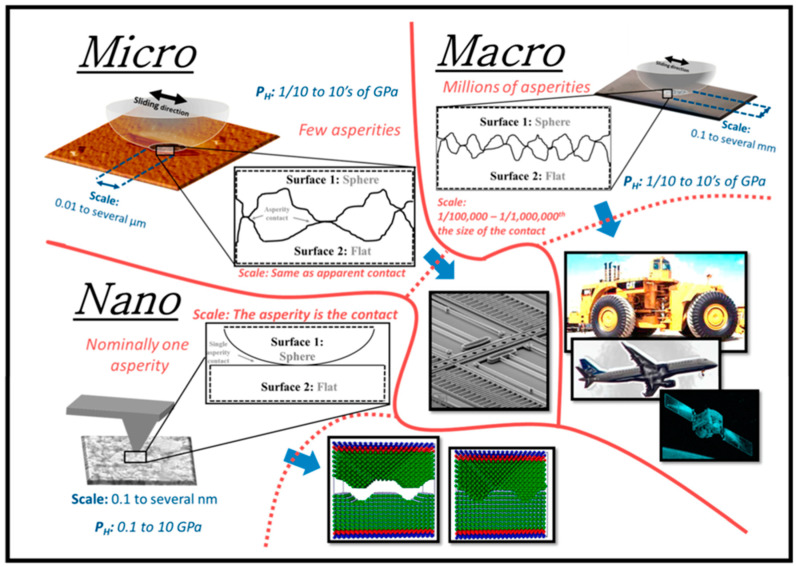
Different length scale (i.e., macro, micro, and nano) investigations used in tribology [127]. This figure is taken from Reference [127] with permission from the publisher.

**Figure 11 materials-13-03489-f011:**
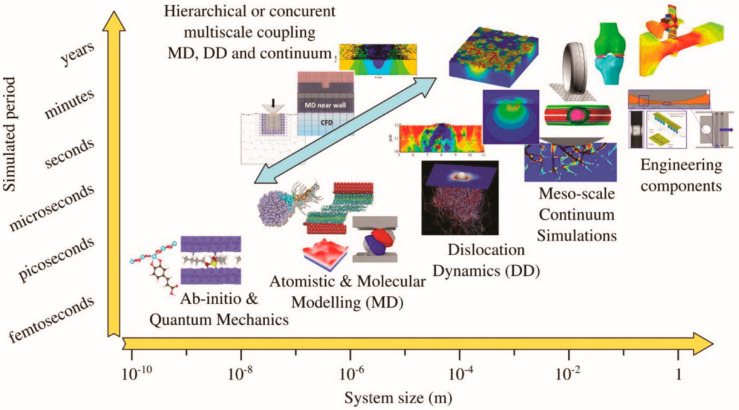
A time versus length scales map of models developed in tribology highlighting the intrinsic link between multiscale/physics [15]. This figure was taken from Reference [15] with due permission from the publisher.

**Figure 12 materials-13-03489-f012:**
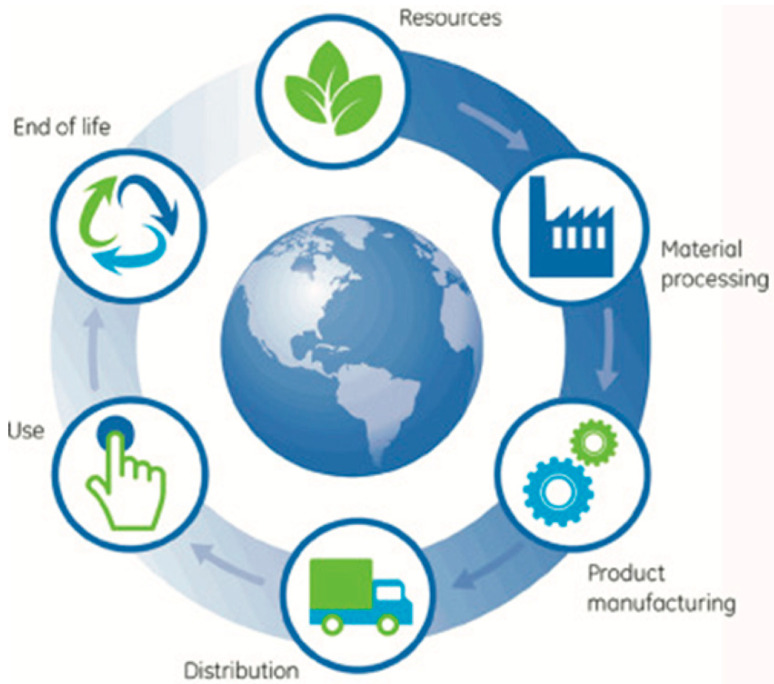
Components of life cycle assessment [153]. This figure was taken from Reference [153] with due permission from the publisher.

**Table 1 materials-13-03489-t001:** Literature survey on simulation methods used to model and predict lubrication effect on tribology.

Simulation Method	Main Features	Authors	Main Findings
Molecular dynamic simulation	Based on calculating the kinematics among particles in molecular level	Harrison et al. [42]	Friction of diamond composite film, diamond-like carbon, amorphous carbon, and self-assembled monolayer systems
Finite elements and boundary conditions method	FEM set explicit relationship between stress and strain and BEM considered the relationship between force and pressure with displacement.	Ducobu et al. [39] Perazzo et al. [59] Rojas et al. [60] Zhang et al. [61] Lian et al. [63] Din et al. [64]	Investigate the effect of geometry on tool wear that was made with Ti6Al4V. Predict abrasive wear on steel against various types of copper. Predict friction on relative wear on mining hopper. Predict fretting fatigue and wear. Model fatigue life and wear on railways. Predicted the damage on fiber reinforced polymers caused by adhesive wear.
Crystal plasticity	Used to model elastic–plastic deformation of metal, in which it is assumed that plastic deformation is a result of plastic slip on metal crystals.	Barbe et al. [72,73] Thamburaja et al. [74] Kalidindi et al. [75] Staroselsky et al. [76]	Modelling elastoplastic phenomenon of polycrystalline aggregates. Model the deformation twinning and martensitic transformation deformations
Discrete dislocation dynamics	Modelling solid material as a linear elastic continuum, and it is used to model the plasticity phenomenon at the micro-scale level.	Fengwei et al. [119] Jinxuan et al. [120]	To model the frictional behavior of a metal asperity on a large single crystal. Simulate the subsurface damages microstructural alteration of titanium alloy.
Solid/fluid interactions and finite volume method	The model is dependents on a solver that deals with solid/fluid interactions by taking into account multiphase phenomena and complex geometry using mass conservation method.	Still in its development stage.	
Machine learning method	Developing a model that finds the complex pattern recognition and regression analysis in a set of data by training the algorithm by given data set without human involvement.	Anand et al. [16] Zakaulla et al. [92] Borjali et al. [93] Xu et al. [94] Gouarir et al. [97] Tran et al. [85] Slavkovic et al. [99] Suresh et al. [102]	Optimization of friction parameters by using a force ANN. Predicted coefficient of friction and wear rate of polycarbonate-based composite by using ANN. Predict the wear rate of polyethylene by linear regression. Data-driven models to improve the fault tolerant ability by using the wear problems in diesel engine. Predict the tool wear by using CNN. Predict wear by using Gaussian process regression Predict the wear rate of iron casting By using SVM Model wear on polymers by using ANN.

**Table 2 materials-13-03489-t002:** Historical hip joint simulator description.

Simulator’s Name	Station	Classification	Motion Simulated	Position and Wear Rate
TE 86 MULTI-STATION- Helsinki University of Technology [141]	Twelve	Two-axis motion	Abduction-adduction (AA): 12 °C Flexion-extension (FE): 46 °C	anatomical
AMTI’s VIVO [142]	Six	Three-axis motion	Abduction-adduction (AA): ±25 °C Flexion-extension (FE): 200° (Cartesian) 110° (Grood and Suntay)	anatomical
AMTI [143]	Twelve	Three-axis motion	Abduction-adduction (AA): ±25 °C Flexion-extension (FE): ±9 °C	Anatomical With wear rate: 4.8 ± 1.1 mg/Mc
Vitro simulations-ProSim pendulum friction simulator (Simulation Solutions Ltd., Stockport, UK) [144]	Single or six	Two-axis motion	Flexion-extension (FE): ±15 °C	anatomical
Bionix^®^12-Station Hip Wear SimulatorMTS systems crop, MN, USA [145]	Twelve	Bi-axial rocking motion	Abduction-adduction (AA): 23 degrees Flexion-extension (FE): 37 °C ± 2 °C	anatomical
SW [146]	Twelve	Two-axis motion	biaxial rocking motion ±23 °C	Non-anatomical With wear rate: 0.17 mg/Mc
SW 2 [147]	Twelve	Two-axis motion	biaxial rocking motion ±23 °C	Non-anatomical With wear rate: 0.032 ± 0.028 mg/Mc
HUT-4 [148]	Twelve	Two-axis motion	Abduction-adduction (AA): 12 degrees Flexion-extension (FE): 46 °C	Anatomical position With rear rate: 8.2 mg/Mc

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
