# Peer review of "Practice of Simulation and Life Cycle Assessment in Tribology—A Review"

_materials, 2020, doi:10.3390/ma13163489_

Round 1

Reviewer 1 Report

This paper presents a review on the existing problems in tribology field, emphasizing the importance of both computer aided simulation and physical experimentation. Also, the paper underlines that the future simulations and life cycle assessment (LCA) of tribological pairs requires a large database of experimental work and machine learning (ML) algorithms and predictive artificial neural networks (ANN).  

The review is rich in information but, in my opinion, it is still difficult to read and to follow a general aim, or a general idea. Unfortunately, there still are some phrases requesting amendments, like those from lines 77-79, 99-101, or 124-127. I advise the authors to carefully analyze the paper, trying to group the similar ideas in the same section, e.g., the description of hip simulator in industrial applications section is not suited. This should be included in the section containing the description of tribometers.

I realize the huge effort made by the authors to bring the paper to publication standards, so I ask for a major revision from the reasons presented above.

Author Response

Lines 77-79, 99-101 were rewritten and lines 124-127 were omitted for better flow of the text. Effort was also given to categorize similar ideas in the same section. Towards that, ‘input and output parameters’ and ‘data acquisition and cleaning data set’ sections were moved after the description of various machine learning methods. Description of hip simulator was omitted, as in the paper there was no section related to description of tribometers. General aim of the paper was also highlighted explicitly. All the changes were made in blue text in the manuscript file to identify them easily.

As the manuscript was revised again in view of reviewers’ valuable comments, the authors hope the manuscript now hold its standard to be published in Materials journal.

Reviewer 2 Report

      This is a resubmitted manuscript reviewing the recent trends of simulation practice in tribology to model tribo-contact scenario and life cycle assessment. The resubmitted manuscript still exists lots of flaws. As indicated in previous comments, many of the discussions are still superficial, which makes the content of this manuscript too long, and boring to the readers. The authors should really consider how to condense the content and add some real in-depth discussion. The focus should be discussions on the new discovering for the recently published papers, as well as the unsolved issues, other than the general description of some old knowledge. A table should be considered to describe the features of different simulation methods for tribology modeling. Meanwhile, all the figures are used to illustrate the general mechanism of the simulation process, it would be better to involve in some simulated results from the published paper, to assist the readers to understand how the methods could be adopted in real research. Therefore, I still suggest rejecting this manuscript, or at least a major revision should receive from the authors.

Author Response

To make the paper more readable, a number of changes were made, such as (i) reorganizing different sections (to keep similar ideas in same the section), (ii) highlighting the general aim of the manuscript, (iii) removing figure (hip simulator), (iv) inclusion of a table that summarizes the main features of different simulation techniques, and (iv) overall better linking between two main sections of the paper that is, simulation and life cycle assessment. All the changes were made in blue text in the manuscript file to identify them easily.

In view of reviewers’ previous comments (‘many of the discussions are still superficial’), the authors response is that, this is how the other review papers were written and presented in literature. Instead of such generalized comments, if the reviewer could point the finger on specific issues, such as which discussion is wrong and how it could be improved further, then the authors are happy to revise that accordingly. For example, by taking consideration of the reviewer’s valuable comment a new table was included that summarizes the main features of the simulation techniques that are commonly used in tribological related applications.

As the manuscript was revised again in view of reviewers’ valuable comments, the authors hope the manuscript now hold its standard to be published in Materials journal.

Reviewer 3 Report

The author summarized the experimental and simulation approaches to tribology study, as well included the life cycle assessment. The main question for this manuscript is the connection between these two topics. For the tribology part, the main discussion focused on the investigation of lubrication performance and tribological contacts. Different simulation approaches were introduced to help readers as references.

While turning to the second part, LCA, the logical connections to the first part was weak. The life cycle assessment related to tribology could be the durability of tribo-materials or oil/additives, recycle of the lubricants, development of eco-friendly lubricants, etc.. It is difficult to find the relationship between LCA to the previous part (mainly on lubrication analysis). It is suggested that the author needs to work on the logic flow of these two parts.

Author Response

In view of reviewers’ suggestion, considerable effort was given to overcome the current limitation of the paper as pointed by the reviewer. The changes include (i) highlighting the importance of LCA in the introduction section (page 9 of current version of the manuscript), (ii) including general aim of the manuscript and overall (iii) better justification of LCA section on pages 51-52 of the current version of the manuscript. All the changes were made in blue text in the manuscript file to identify them easily.

As the manuscript was revised again in view of reviewers’ valuable comments, the authors hope the manuscript now hold its standard to be published in Materials journal.

Round 2

Reviewer 1 Report

I suggest to the authors to compress Tables 1 and 2, eventually decreasing the text size. Also, Notations and symbols and Table of contents should be placed at the end of the paper. Finally, I congratulate the authors for their work.

Author Response

Tables 1 and 2 were compressed by decreasing the text size (lines 524-525 and 691-692). Notations and symbols and Table of contents were placed at the end of the paper (lines 25-62). All the changes were highlighted in yellow colour.

Reviewer 2 Report

The reviewer was satisfied with the new version. 

Author Response

Thanks for reviewers’ comments.

Reviewer 3 Report

The author made explanations in detail. The importance of the tribology part and LCA are both discussed. However, as motioned in the previous report, both parts are well discussed; but the logical flow of the manuscript is missing. Less effort has been put into this problem. It is suggested that the author should focus on one topic only and resubmit.

Author Response

There are a number of papers available in literature that deals with the simulation of tribology and LCA, separately. Thus, one of the main objectives of the present paper is it combine these two aspects and made the link in terms of how simulation of tribology can contribute towards LCA together with experimental (lines 127-155 and 722-730). As the review pointed out, detail explanation was made together with the importance of each aspects. Substantial link between these two aspects were made (lines 127-155) with clear set up objective in the introduction section. As mentioned by the authors previously, if the reviewer specifically suggests where it is missing logical flow then the authors are happy to consider that in their revision.

In addition to that, all other reviewers have accepted the paper in its present form and it’s the authors’ intention also to keep both parts, which is the novelty of the present paper.